# The Involvement of Alarmins in the Pathogenesis of Sjögren’s Syndrome

**DOI:** 10.3390/ijms23105671

**Published:** 2022-05-18

**Authors:** Julie Sarrand, Laurie Baglione, Dorian Parisis, Muhammad Soyfoo

**Affiliations:** 1Department of Rheumatology, Hôpital Erasme, Université Libre de Bruxelles, 1070 Brussels, Belgium; julie.sarrand@ulb.be (J.S.); laurie.baglione@ulb.be (L.B.); dorian.parisis@ulb.be (D.P.); 2Laboratory of Pathophysiological and Nutritional Biochemistry, Université Libre de Bruxelles, 1070 Brussels, Belgium

**Keywords:** Sjögren’s syndrome, alarmins, damage associated molecular pattern, cytokines, innate immunity, autoimmunity

## Abstract

Sjögren’s syndrome (SS) is a chronic autoimmune disease that affects exocrine glands, primarily the salivary and lachrymal glands. It is characterized by lymphoplasmacytic infiltration of the glandular tissues, ultimately leading to their dysfunction and destruction. Besides classic dry eyes and dry mouth defined as sicca syndrome, patients affected by the disease also typically display symptoms such as fatigue, pain and in more than 50% of cases, systemic manifestations such as arthritis, interstitial lung involvement, neurological involvement and an increased risk of lymphoma. The pathophysiological mechanisms underlying SS still remain elusive. The crucial role of innate immunity has been advocated in recent years regarding the pathogenesis of pSS, especially in the initiation and progression toward autoimmunity. Alarmins are endogenous molecules that belong to the large family of damage associated molecular pattern (DAMP). Alarmins are rapidly released, ensuing cell injury and interacting with pattern recognition receptors (PRR) such as toll-like receptors (TLR) to recruit and activate cells of the innate immune system and to promote adaptive immunity responses. This review highlights the current knowledge of various alarmins and their role in the pathogenesis of pSS.

## 1. Introduction

Sjögren’s syndrome (SS) is a chronic systemic autoimmune disease characterized by lympho-plasmacytic infiltration of the exocrine glands responsible for xerostomia and keratoconjunctivitis sicca, also referred to as sicca syndrome [1]. The disease is considered as primary SS (pSS) if it occurs in the absence of other autoimmune diseases, or as secondary SS (sSS) if it is associated with other autoimmune disorders such as systemic lupus erythematosus (SLE), rheumatoid arthritis (RA) and systemic sclerosis (SSc) [2,3]. The disease symptoms often stretch beyond the classical sicca syndrome and include systemic manifestations such as arthritis, interstitial lung involvement, neurological involvement and even an increased risk of lymphoma [4,5].

There has been a quantum leap in recent years to unravel the pathophysiological mechanisms underlying SS, but the exact triggers of the disease still remain unknown. Several converging lines of evidence indicate that innate immunity plays a primary role in the pathogenesis of pSS, especially in the initiation and progression toward autoimmunity [6]. The conjuncture of a genetic predisposal, environmental insults and hormonal disequilibrium may lead to the activation of the resting epithelium, the up-regulation of toll-like receptors (TLR) such as TLR-2, 3, 4, 7, 8, 9 [7,8,9,10], leading to the liberation of damage associated molecular pattern molecules (DAMP) and pro-inflammatory cytokines such as interferon (IFN), tumor necrosis factor (TNF)-α and interleukin (IL)-6, IL-7, IL-17, further stimulating downstream inflammation [9,11]. The subsequent activation of plasmacytoid dendritic cells (pDC) triggers a massive release of IFN-α and the release of B cell activating factor (BAFF) that support the proliferation of B cells inside germinal centers (GC). IFN-γ in turn acts on the salivary gland epithelial cells (SGEC) to further release alarmins in the extracellular milieu, thereby constituting a vicious auto-inflammatory loop fostering local and systemic damage and contributing to disease perpetuation [12].

Alarmins are endogenous and constitutive molecules that belong to the large family of damage associated molecular pattern (DAMP). Ensuing degranulation, cell injury, cell death or immune cells activation [13,14,15], alarmins are released into the extracellular space where they mediate intercellular signals through interactions with chemotactic and pattern recognition receptors (PRRs) such as TLR and receptors for advanced glycation end products (RAGE), further eliciting innate immunity [13,14,16]. Additionally, alarmins have the ability to prompt adaptive immunity responses and T cell-dependent long-term immune memory through their capacity to induce DCs maturation [15]. Any nucleated cell can be a potential source of alarmins and can be found in distinct compartments of the cell such as the nucleus, cell granules, vesicles, cell membrane or in the cytosol.

This review depicts the different types of alarmins that have been reported to be dysregulated in pSS patients and highlights the current knowledge about their potential contribution to the pathophysiology of the disease. For purposes of clarity, we have articulated this review according to the localization of the alarmins in the cell compartment (Table 1).

## 2. Nuclear Alarmins in SS

### 2.1. HMGB1

High Mobility Group Box 1 (HMGB1) is a non-histone nuclear protein that is actively secreted by activated immune cells or passively released by injured or dying cells and acts as a pro-inflammatory mediator by binding to various receptors on the surface of responding cells, such as RAGE, TLR-2, TLR-4 and TLR-9 [17,18].

Significantly increased expressions of HMGB1 were identified in salivary glands (SG) of pSS patients suggesting that HMGB1 might play a role in the etiology of SS [19,20]. In 2006, Ek et al. found an increase in extracellular HMGB-1 in the peri-lymphocytic area of the minor SG of patients with SS compared to patients with sicca symptoms alone or healthy controls. Furthermore, TNF-α and IL-1 levels are higher in patients with SS than in control subjects. Secreted TNF-α and IL-1 in patients with SS are localized in the same area as extracellular HMGB-1 in the lymphocytic infiltrates, suggesting that HMGB-1, together with TNF-α and IL-1, may constitute a pro-inflammatory loop in SS, promoting persistent glandular inflammation [19]. In 2012, Dupire et al. found that serum HMGB1 levels were significantly higher in SS patients with anti-SSA and active disease [21]. However, they did not observe any correlation of serum HMGB1 levels with EULAR Sjögren’s Syndrome Disease Activity Index (ESSDAI) [20]. Conversely, in 2017, Kanne et al. proposed that total HMGB1 serum levels are also linked to disease activity, as measured by the ESSDAI. They described higher HMGB1 levels in individuals with extra-glandular disease manifestations compared to patients without extraglandular involvement [22]. In SS patients with high disease activity, HMGB1, as well as soluble RAGE (sRAGE) levels, are elevated, suggesting that HMGB1 and its counterpart sRAGE are involved in the etiology of SS, and that the HMGB1/RAGE axis could be a therapeutic target [22] (Figure 1).

In SS, due to the inflammatory milieu, the conjunctival, corneal, and lacrimal epithelial cells are injured, and some of them may necrotize, releasing extracellular HMGB1. Extracellular HMGB1 in SS may induce significant auto-inflammatory cycles by triggering an adaptive immune response, resulting in a persistent pathological state [23]. According to the study of Kim et al., sub-conjunctival administration of anti-HMGB1 attenuates the clinical manifestations of dry eye [23].

In 2020, Di Wang et al. reported that levels of HMGB1 and aquaporin 5 (AQP5) are, respectively, increased and decreased in SS-triggered xerostomia. Xerostomia was reversed in mice treated with anti-HMGB1 due to increased saliva production and lower anti-SSB levels. Inhibiting HMGB1 reduces SS-induced xerostomia by inhibiting the HMGB1/TLR4/nuclear factor-kappa B (NF-κB) signaling pathway and increasing AQP5 expression [24]. This could potentially be a new therapeutic strategy for the treatment of xerostomia in SS.

### 2.2. IL-1α

IL-1α is a dual-function cytokine, meaning that in addition to its extracellular receptor-mediated activities as a classical cytokine, it also has a transcriptional role [25]. IL-1*α* binds to the IL-1 receptor 1 (IL-1R1), as does IL-1β, and has the same pro-inflammatory effects [26]. IL-1*α* is a primary mediator of local inflammation, influencing the local microenvironment either by exposure on the membranes of stressed cells followed by juxtacrine binding of IL-1R1 on adjacent cells, or through release upon cell death followed by the paracrine binding of IL-1R1 on nearby cells [26].

A higher concentration of IL-1*α* was found in the salivary fluid and blood of patients with SS in several studies focusing on cytokine profiles that may contribute to the pathological scenery of this disease [27]. In a previous study from 1998, several cytokines, including IL-1α, have been found in SS patients’ biopsies, with cytokine mRNA found in both infiltrating lymphocytes and SGEC. IL-1 gene expression was only found in SS patients and not in the chronic sialadenitis controls [28]. Fox et al. also reported that SGEC generated more than 40-fold more IL-1*α* mRNA than epithelial cells from people with histologically normal SG [29]. Experimental studies in mice showed that direct instillation of IL-1α in SG [30] induces inflammation, dysfunction and loss of acinar epithelial cells [26]. IL-1 receptor antagonist (IL-1Ra) levels in the saliva of SS patients have been demonstrated to be substantially lower than those of normal controls [31], suggesting that IL-1 produced by epithelial cells rather than immune cells influences the onset and progression of SS in the target organs [32]. Dubost et al. indicated that an imbalance in salivary IL-1/IL-Ra may cause inflammatory lesions in the mouth and impair mucosal cell differentiation in SS [33] (Figure 1).

The equilibrium of cytokines in tear fluid and the conjunctival epithelium is altered in SS [34]. IL-1*α* is expressed at higher levels in the conjunctival epithelium of SS patients. In 1998, Jones et al. hypothesized that a lack of aqueous tears produces mechanical abrasion, which encourages conjunctival epithelial cells and lymphocytes to produce and release IL-1 and various cytokines into the tear film [35]. In 1999, Pflugfelder et al. found that patients with SS had significantly higher levels of IL-1α RNA transcripts in their conjunctival epithelium than controls, but lower tear fluid epidermal growth factor (EGF) concentrations. IL-1α RNA in the conjunctival epithelium increased as Schirmer scores decreased. As tear fluid EGF concentration drops and levels of inflammatory cytokines in the conjunctival epithelium rise, the severity of keratoconjunctivitis sicca worsens [34].

In 2001, Solomon et al. hypothesized that increased concentration and activity of IL-1 could be an initiating factor for the observed ocular surface immunopathology of dry eye. They found that IL-1*α* and mature IL-1β concentration in the tear fluid was increased in patients with SS aqueous tear deficiency when compared to controls. The intensity of corneal fluorescein staining was shown to have a substantial positive association with tear fluid IL-1*α* concentration. In tear fluid, dry eye illness is associated with an increase in the proinflammatory forms of IL-1 (IL-1*α* and mature IL-1β) and a reduction in the physiologically inactive precursor IL-1β [36].

Furthermore, increased expression of IL-1 has been linked to SS-associated keratinizing squamous metaplasia of the ocular mucosal epithelium [37,38], indicating that IL-1R targeted treatment may be beneficial for treating SS-associated ocular surface illness, including dry eye [32]. Because IL-1 appears to be directly involved in the genesis of SS, an anti-IL-1 medication might be used as a possible therapy [32]. A recent prospective double blind randomized trial showed that targeting IL-1 with anakinra in a topical application is beneficial in reducing dry eye disease-related symptoms and corneal damage [27,39]. Topical IL-1 inhibitory actions may be more useful for SS patients in the future than disease-modifying antirheumatic drugs (DMARDs) or biological response modifiers [32].

Because the acinar cells, duct cells and blood vessels of the lacrimal glands are innervated by the sympathetic and parasympathetic nervous systems, numerous studies have found that exogenously administered IL-1 might inhibit neurotransmitter release [27,40,41,42]. Pro-inflammatory cytokines reduce both neurally and adrenergic agonist-induced lacrimal gland protein production. High levels of IL-1, as seen in SS exocrine glands, may compromise the secretory function of these organs [42].

Several chronic inflammatory diseases, including SLE and RA, have been associated with a higher risk of venous thromboembolism. However, the data on SS remain limited. In 2015, a study showed that the kinetics of blood clot formation and platelet aggregation in pSS is normal and that IL-1*α* and other cytokines (macrophage inflammatory protein (MIP)-1, IL-17A, IL-1, and IFN-γ) may correlate with clot strength and overall coagulability in pSS [43].

### 2.3. IL-33

IL-33 is an epithelial-derived pro-inflammatory alarmin with nuclear localization and chromatin binding [44]. Under normal tissue homeostasis circumstances, intracellular IL-33 acts as a guardian, protecting the integrity of barrier cells through gene expression control. After cell damage, necrosis or injury, IL-33 is rapidly released into the extracellular space where it binds to its cognate receptor suppression of tumorigenicity 2 (ST2) and to its co-receptor IL-1RAcP.

Previous research has suggested that IL-33/ST2 may play an essential role in autoimmune disorders such as SS or SLE and RA [45]. In pSS patients, blood levels of IL-33 and ST2 have been reported to be considerably higher than in controls. We have shown that IL-33 expression was higher in pSS patients with Chisholm scores (CS) of 2 and 3, and then a decrease in CS of 4. Jung et al. also reported that SG with greater histological grade exhibited a decreasing pattern of IL-33 and ST2 expression as the degree of inflammation progressed. In pSS, IL-33 alone does not induce the release of pro-inflammatory cytokines [12].

A recent study proposed a hypothesis concerning the involvement of the IL-33/ST2 axis in the pathogenesis of pSS. There is an upregulation of TLR-3, -4 and -7 and subsequent activation of the salivary epithelium in the presence of genetic predisposition, hormonal imbalance or environmental stress. In the early stages of the disease, the injured epithelium increases the expression of IL-33 in an attempt to suppress autoimmune dysregulation. IL-33 is released in the extracellular space as the disease progresses and in advanced stages in response to pro-inflammatory stimuli and additional epithelium damage. Counter-regulatory mechanisms are activated to restrict IL-33’s pro-inflammatory activities, such as an increase in the IL-33 decoy receptor soluble ST2 (sST2). Furthermore, IL-33 works in synergy with IL-12 and IL-23 on natural killer (NK) and NK T cells to boost IFN production, which contributes to inflammation. TNF-α, IL-1 and IFN, in turn, promote the activation of the IL-33/ST2 pathway, forming a vicious auto-inflammatory cycle that causes local and systemic damage and contributes to illness persistence [12] (Figure 1).

Zhao et al. were the first to report increased sera levels of both IL-33 and ST2 in pSS patients with interstitial lung involvement (ILD). Based on a previous study [46] showing that IL-33 may cause significant histological alterations in the lungs of mice, they hypothesized that IL-33 may stimulate the production of IL-6 and IL-8, resulting in lung injury in individuals with pSS [47]. No association between IL-33 levels and biological markers (such as C-reactive protein (CRP), disease duration, Chisholm score (CS) or ESSDAI was reported. These findings are in contrast with the study of Jung et al., where sST2 levels among pSS patients were linked to ESSDAI, but also thrombocytopenia or disease duration [45]. Since the concentration of IL-33/sST2 appears to be associated with disease activity, this suggests that it could be used as a biomarker [48,49,50,51,52]. Moreover, this study confirmed the expression of IL-33 in epithelial and endothelial cells of glandular tissue in individuals with pSS and revealed that IFN-γ stimulation can increase the synthesis of IL-33 mRNA by SGEC [45].

The role of IL-33 in the ocular severity of pSS patients was recently demonstrated. In comparison to non-SS dry eye patients, levels of IL-33 in the tears of pSS patients were considerably higher [53,54]. Elevated tear levels of IL-33 were positively correlated with symptom scores but negatively correlated with tear film breakup time and Schirmer test in both non-SS dry eye and SS dry eye patients [53]. IL-33 may contribute to disease severity by promoting type 2 helper cells (Th2) inflammation in the development of dry eye [53].

## 3. Cytoplasmic Alarmins in SS

### 3.1. Heat Shock Proteins

Heat shock proteins (HSP) comprise a family of proteins that is highly conserved across cellular evolution and are classified according to their molecular weight. Under physiological circumstances, they play a key role as intracellular chaperones by assisting the correct folding or the clearance of misfolded proteins during protein synthesis. In response to stress, some members of this family, namely HSP60, HSP70 and HSP90, are released into the extracellular space [55,56] where they recognize PRR such as TLR-2 and TLR-4, leading to activation of the pro-inflammatory mitogen-activated protein kinase (MAPK) cascade and the NF-κB signaling pathway [57,58]. Extra-cellular HSP can also bind to cell surface glycoproteins that act as scavenger receptors on dendritic cells (cluster of differentiation (CD40, CD91), macrophages (CD14, CD40, CD91), monocytes and B cells (CD24, CD40) [59,60,61,62], further eliciting pro-inflammatory responses. HSP can also be captured by surrounding antigen-presenting cells (APC), allowing cross-presentation of the peptides by major histocompatibility complex (MHC) class I that further contributes to the progression toward autoimmunity [63].

The first study reporting involvement of HSP in SS dates back to 1991 when they found immunoglobulin (Ig)G antibodies against HSP90 and HSP60 and IgM antibodies against HSP60 and HSP73 [64]. Several other studies confirmed the increased rate of antibodies against HSP60 [65,66,67]. Furthermore, increased HSP60 expression was also demonstrated in 10 labial SG of pSS patients compared to 10 non Sjögren sicca patients. IgG antibodies against HSP60 significantly correlated with focus score (FS) and systemic disease parameters such as ESR, serum total IgG [65]. Conversely, one study found lower levels of anti-HSP60 and anti-HSP65 antibodies in patients compared to in controls [68]. Taken together, the present studies suggest a potential contribution of HSP60 to immunopathology in pSS. Whether HSP60 triggers the immune response or whether HSP60 titers are secondary to another inflammatory trigger remains unclear and has to be studied in more detail. In addition, the immunization with HSP60 in an experimental pSS model inhibited SS symptoms [69]. Moreover, several human HSP60-derived peptides induced regulatory T-cell responses in arthritis models [70,71]. These HSP60-derived epitopes could therefore be used for epitope specific immunosuppression. Future studies need to document the type of immune responses that are induced by these peptides and how these modulate immune responses in pSS (Figure 2). Another study reported the presence of IgG antibodies against HSP65 in 85.7% of pSS patients and in 2.8% of healthy controls [72]. Another study found both HSP47 levels and IgG antibody against HSP47 significantly higher in pSS patients compared to healthy controls [73].

Plasma concentrations of HSP90 were significantly higher in pSS patients with high fatigue compared with those with low fatigue, and there was a tendency to higher concentrations of HSP72 in patients with high fatigue compared to patients with low fatigue. Thus, extracellular HSPs, particularly HSP90a, may signal fatigue in chronic inflammation [74].

HSP antibodies are not specifically found in pSS but seem to be found in a variety of auto-immune disease. Raised levels of circulating anti HSP65 and HSP60 antibodies were found in patients with RA, SSc, Behçet and psoriasis [75,76,77] (1418288, 8423405, 2004821). HSP47 were also found in patients with RA, SLE and mixed connective tissue disease (MCTD) [73].

### 3.2. S100 Proteins

S100 proteins are a family of low molecular weight calcium-binding proteins that are characterized by a large variety of intra-cellular and extra-cellular functions. They exist as homodimers, heterodimers or multimers. When S100 proteins are released in the extracellular compartment, they can exercise their alarmin function through their liaison to receptors such as RAGE [78] and TLR-4 [79], triggering an inflammatory response, prompting both innate and adaptive immune responses [80,81].

S100A8 and S100A9, respectively known as myeloid related protein (MRP) 8 and MRP14 [82,83], mainly exist as a heterodimer, S100A8/S100A9 (also termed calprotectin). Calprotectin is massively produced by activated granulocytes and MC in a wide range of inflammatory conditions such as RA and SLE, and even correlated with disease activity [84,85,86]. S100 A8/A9 is up-regulated in both serum and SG of pSS patients [87,88,89,90] and even correlated with FS [87]. In another study, however, serum S100A8A/A9 levels did not differ significantly from healthy patients, while salivary S100A8/A9 levels were significantly increased in SS patients [91]. Moreover, fecal calprotectin was significantly higher in pSS compared to healthy controls and correlated with ESSDAI [92,93,94]. However, calprotectin was higher in pSS with organic gastro-intestinal disease compared to the other patients with pSS [92]. A proteomic analysis of parotid saliva suggested that levels of S100A8 and S100A9 might act as a biomarker for the development of mucosa-associated lymphoid tissue (MALT) lymphoma. Indeed, it allowed for discrimination between SS patients and healthy controls and between patients with SS MALT lymphoma and SS without lymphoma [95]. Another study showed a positive correlation of S100A8/A9 between parotid salivary levels and minor SG FS of SS patients [89].

S100B is abundant in the central nervous system and is produced more particularly by activated astrocytes in response to homeostatic disturbances [96]. S100B was increased in the cerebrospinal fluid (CSF) of pSS patients [97,98,99]. The proposed hypothesis was that activated astrocytes would signal through RAGE and TLR-4 on microglia and lead to IL-1β production [100,101]. The activation of astrocytes and microglia may therefore be the initial step in fatigue generation.

Regarding the expression of TLR-4 in pSS patients, a higher expressional level of TLR-4 was demonstrated in both SGEC and salivary-infiltrating mononuclear cells relative to controls [102]. Moreover, functional studies in cultured human salivary cells suggested that activated TLR-4 stimulate CD54 expression and IL-6 production through the phosphorylation of MAPK, further incriminating TLR-4 in the pathogenesis of pSS [6,8]. A study determined RAGE expression in the labial SG of pSS patients and showed over-expression of RAGE in pSS tissues compared to healthy controls [103] (Figure 2).

### 3.3. ATP

Adenosine triphosphate (ATP) is a nucleotide belonging to the purine family that plays a key role in the metabolism of cells by conveying energy. When released into the extracellular space, it initiates chronic inflammatory responses through the activation of P2 purinergic receptors such as P2X and P2Y receptors. P2X receptors are ATP-gated ion channels that are specific for ATP, whereas P2Y are G protein-coupled receptors that are specific for adenosine diphosphate (ADP), uridine triphosphate (UTP) and ATP [104,105]. Stimulation by ATP through the P2 purinoceptors of macrophages (MC) and dendritic cells (DC) results in the production of inflammatory cytokines, chemokines and leukotrienes [106,107]. In addition, ATP promotes (Nod-like receptor protein 3) NLRP3 inflammasome assembly and activation through P2X7 [108].

Studies using labial SG biopsies from SS patients demonstrated an increased expression of P2X7R, caspase-1, IL-1β, IL-18 and NLRP3 that positively correlated with SG FS [109,110]. Studies conducted in mice confirmed that ATP signaling through the P2X7 receptor (P2X7R) contributes to SG inflammation by promoting apoptosis of SGEC, caspase activation and stimulation of the NLRP3 inflammasome, leading to the secretion of pro-inflammatory cytokines such as IL-1β and IL-18 [111,112]. In addition, a prospective study highlighted that P2X7R expression may be a useful biomarker for MALT lymphoma development as P2X7R expression in labial SG of SS was significantly higher in patients who will further develop MALT lymphoma in comparison to SS patients who will not [110]. Another group demonstrated a higher expression of P2X7R on peripheral blood mononuclear cells (PBMC) from patients with pSS compared to rheumatoid arthritis patients and healthy controls [113]. Antagonism of the P2X7R has been investigated as a potential treatment for several inflammatory diseases, including SLE [114,115] and RA [116]. Preclinical studies reported that pharmacological antagonist of the P2X7R significantly reduced sialadenitis and improved salivation in a murine model of SS [111]. Another study performed with PBMC of pSS patients demonstrated that paeoniflorin significantly inhibited the ATP-induced expression of P2X7R and down-regulated the production of IL-1β and IL-6 [117]. Taken together, these studies suggest that P2X7R inhibition could be a promising tool for targeting SG inflammation (Figure 2).

The P2Y2 receptor (P2Y2R) is not expressed in the SG under physiological conditions but is up-regulated in the major SG of several mouse models of pSS [118,119]. Functional studies on mice incriminated P2Y2R signaling in the recruitment, migration and proliferation of immune cells [120,121]. Besides, genetic ablation of P2Y2R attenuates sialadenitis in a mouse model of SS [119], suggesting that targeting the P2Y2R could constitute another effective therapeutic approach to reduce SG inflammation (Figure 2).

### 3.4. Uric Acid

Uric acid (UA) is the end metabolism product of the purine pathway. When released in the extracellular space, it forms monosodium urate (MSU) microcrystals. It was not until recently that extracellular uric acid was identified as an alarmin through its multiple interactions with immune cells. UA is a potent inducer of dendritic cell maturation, subsequently prompting responses from CD8+ T cell [122]. The phagocytosis of UA crystals by MC induces the release of proteases such as cathepsin B, which in turn activate the NLRP3 inflammasome and the production of IL-1β and IL-18 [123,124,125,126]. In addition, UA crystals can trigger the mast cells production of nitride oxide, IL-1β, and IL-6 [127,128] and act as a chemoattractant for eosinophils [129] (Figure 2).

There is no study specifically investigating the role of UA in the pathogenesis of SS. A study mentioned the presence of uric acid in both the saliva and serum of SS patients [130]. Another study showed a decrease of uric acid in salivary samples of pSS compared to healthy controls [131].

### 3.5. Galectin-3

Galectin-3 (Gal-3) belongs to the β-galactoside-binding lectine family and is widely expressed in myeloid cells, fibroblast, epithelial and endothelial cells [132,133,134,135]. It can be found in cytosol, as well as extracellularly, after being released following physiological and pathological stimuli [136,137]. Gal-3 has been implicated in human autoimmune diseases (i.e., SLE, Behçet, SSc) and tissue fibrosis (i.e., pulmonary fibrosis) [132,138,139,140]. Gal-3 promotes DC activation and T cell-mediated immune responses [132], leading to pro-inflammatory cytokine production such as IL-6, C-X-C motif chemokine type 8 (CXCL8) and matrix metalloproteinase (MMP)-3 [135,141].

Zhang R. et al. found significantly higher levels of Gal-3 in sera of pSS compared to healthy controls. Serum Gal-3 positively correlated with the development of ILD, CRP, IgG and IL-17. Taken together, this data suggest that Gal-3 could positively regulate the development of type 17 helper cells (Th17) and the production of IL-17, therefore contributing to the development of SS [142] (Figure 2).

### 3.6. Calreticulin

Calreticulin is a calcium-binding protein that is essential for the quality control of protein folding and intracellular calcium homeostasis [143]. Extracellular calreticulin induce the maturation of DC and the expression of pro-inflammatory cytokines through the NF-κB pathway [143]. When completed with peptides, calreticulin can prime specific CD8+ responses against the peptides.

In pSS, current data suggest that calreticulin may play an active role in the generation of an autoimmune response against Ro60. When released into the extracellular space, calreticulin could bind Ro peptides after apoptosis [144], necrosis [145] or cell lysis by cytotoxic T-cells [146], producing conformational changes and thereby increasing their antigenicity. The complex is then transported to professional APC and the peptides are presented to autoreactive T cells [147] (Figure 2).

### 3.7. Thymosin Beta 4

Thymosin beta 4 (Tβ_4_) is a member of beta-thymosin family. Intracellular Tβ_4_ is mainly found in the cytosol and plays a pivotal role in the cell cytoskeletal system, acting as G-actin-sequestering molecules [148]. When released into the extracellular compartment, Tβ_4_ is involved in many critical biological processes, including angiogenesis [149], wound healing [150], inflammatory response [151] and cell migration [152]. In gingival fibroblasts, Tβ_4_ was able to suppress the apoptosis and the production of interleukin-8 following stimulation by tumor necrosis factor-α, acting as an anti-inflammatory and antiapoptotic factor [153]. Though it is believed that Tβ_4_ could act as an alarmin through a yet unidentified receptor, the precise mechanism by which Tβ_4_ regulates the immune response remains unknown (Figure 2).

In the saliva of pSS patients, Tβ_4_ levels were statistically higher with respect to healthy controls and patients with sicca syndrome and other autoimmune diseases. Regarding thymosin immunostaining, Tβ_4_ immunoreactivity was completely absent in minor SG samples of pSS patients [154]. This finding suggests that acinar cells of SG constantly exposed to inflammatory damage of SS might release high amounts of Tβ_4_ in the saliva, where it would exert a cytoprotective effect in tissue repair and in the anti-inflammatory response, as previously shown in the damaged cornea [155]. Alternatively, the increased apoptosis of epithelial cells that characterize pSS can release the peptide, with the new and regenerating epithelial cells not producing enough peptide to be detected by immunohistochemistry [156].

### 3.8. Ribonuclease 7

Ribonuclease 7 (RNase 7) is a ribonuclease that belongs to the RNase A superfamily. It is abundantly expressed in keratinocytes [157] and has also been described in oral epithelial cells [158,159]. Functional in vitro assays showed that RNase 7 in a mixture with self-deoxyribonucleic acid (DNA) was a strong trigger of IFN-α production by human DC through TLR-9 stimulation [160] and a potent repressor of Th2 cytokines production (IL-4, IL-13 and IL-5) by activated Th2 cells and CD4+ T cells [161].

In SS patients, RNase 7 expression was strongly associated with areas of lymphocytic infiltration in minor SG [162]. One could hypothesize that RNase 7, together with self-DNA released by apoptotic SGEC, could contribute to the IFN-α production by DC that is a key feature of the pathogenesis of pSS. Further studies are required to elucidate the exact contribution of RNase 7 in this process (Figure 2).

### 3.9. Cathelicidin (LL37)

LL37—also known as cathelicidin—is a 37 amino acid cationic protein cleaved from the human cationic antimicrobial peptide 18 (cCAP18). LL37 is expressed in murine and human salivary glands, especially at the level of ductal cells [163,164,165]. This expression by ductal cells is increased in patients with chronic sialadenitis [163] or Sjögren’s syndrome [166]. In the latter case, the increase in expression is found in the vicinity of lymphocyte foci and ectopic germinal centers [166].

This peptide is known to form tetramers via its C terminal portion and thus generate pores in bacterial anionic membranes [167]. Additionally, LL37 have different alarmin-like functions for the purpose of innate immunity by stimulating the chemotaxis of immune cells, the function of dendritic cells and by complexing with double-stranded ribonucleic acid (dsRNA) and DNA for liaison to TLR-3, TRL-7 or TLR-9 [167]. In Sjögren’s syndrome, aggregates of apoptotic DNA complexed to LL37 have been observed by immunofluorescence [166]. These complexes stabilize apoptotic DNA and allow it to significantly activate the maturation of B-cells via TLR-9 [166]. Due to its rein-forcing effet on the immunostimulating nature of nucleic acids, LL37 could play a role in the pathogenesis of Sjögren’s syndrome by overstimulating B-cells.

## 4. Granule-Derived Alarmins in SS

### Defensins

Defensins are small cystein-rich cationic proteins that are classified into α- and β-defensins. α-defensins are stored in the granules of neutrophils [168] and intestinal Paneth cells [169], while β-defensins are mainly expressed in epithelial cells [170]. α-defensins exert a chemotactic activity towards monocytes, T-cells and DC [171,172], leading to the production of cytokines by epithelial cells [173] and the enhancement of systemic antibody response [174], which may therefore represent a link between innate and adaptive immunity [175]. β-defensins are involved in antimicrobial activity and display a chemotactic activity for DC and T-cells [176].

α-defensin-1 was identified in tear fluid, saliva samples, the minor SG of pSS patients and was absent in healthy controls [177,178,179]. In addition, the presence of α-defensins-1-3 was also identified in the saliva of pSS patients [177]. A functional study showed that α-defensin-1 is a potent inducer of the type I IFN response genes [180], which plays a key role in the pathogenesis of pSS and are also found in the minor SG from pSS patients [181,182]. This suggests an important role for α-defensin-1 in the pathology of SS by contributing to IFN response (Figure 3). There is more conflicting data regarding the presence of β-defensin in pSS patients. Kawasaki et al. reported a significant up-regulation of the β-defensin 2-gene expression in conjunctival epithelial cells from patients with primary SS [183]. β-defensin 2 was also identifiable in the majority of saliva specimens of patients with primary SS [177], while another study found the expression of human β-defensins 1 and 2 was decreased in SG affected by SS in comparison to the human β-defensin expression patterns in SG from normal subjects [184].

## 5. Plasma Membrane Alarmins in SS

### Annexins

Annexins are a family of 12 calcium-binding proteins involved in a variety of biological processes, including vesicle trafficking, cell division, cell growth regulation, calcium signaling and apoptosis [185]. Antibodies against annexin I, II, V and XI have been described in auto-immune rheumatic diseases [185].

The overexpression of annexin A2 in parotid tissues was found in pSS and pSS with mucosal-associated lymphoid tissue lymphoma (pSS/MALT). According to the western blotting results, this protein was gradually up-regulated throughout the evolution of pSS to pSS/MALT [186]. These findings indicate that, in addition to contributing to the pathophysiology of SS, annexin probably also plays an important role in the progression of pSS to pSS/MALT [186]. Increased levels of annexin-A2 were also observed in pSS saliva sub-proteome, suggesting that saliva might be used as a potential indicator for disease progression and lympho-proliferation in patients with pSS [187] (Figure 3).

Anti-annexin V antibodies are increased in people with fibromyalgia syndrome (FMS), especially those who have SS. Annexin, in addition to the other recognized essential participants in the apoptotic process in SS, may be one of the most crucial variables, particularly in those with FMS [188].

Anti-annexin XI antibodies may enhance the vulnerability to inflammatory or autoimmune disorders by impairing annexin XI’s ability to regulate apoptosis. These antibodies were found in the sera of patients with RA, pSS and SLE, particularly those with anti-SSB antibodies [189]. So far, no definitive correlations between anti-annexin antibodies and clinical or serologic characteristics have been discovered [185].

## 6. Extra-Cellular Matrix Alarmins in SS

### Decorin

Decorin (DCN) are proteoglycans that belong to the small leucine-rich proteoglycan family. Beside its structural role as a component of the extracellular matrix, it is involved in different biological processes such as inflammation, proliferation and migration [190,191]. DCN contribute to the development of a pro-inflammatory environment by binding both TLR-2 and TLR-4 on MC, leading to the activation of the MAPK and NF-κB signaling pathways and the production of cytokines such as TNF-α and IL-12 [192,193,194] and programmed cell death 4 (PDCD4), an inhibitor of the transcription of the anti-inflammatory cytokine IL-10 [193]. In addition, DCN also stimulates chemokine (C-C motif) ligand 2 (CCL2) production, a potent chemoattractant for mononuclear cells [195].

In pSS, the role of DCN has not yet been studied. However, in a mouse model of SS (NOD.B10), promising results indicated that DCN signal through TLR-4, but not TLR-2, and induce TNF-α in splenic tissue [196]. Further studies are needed to determine whether DCN released from injured SG op pSS could bind TLR expressed by SGECs and tissue resident immune cells, thereby driving the production of pro-inflammatory TNF-α and contributing to the migration of innate and adaptive cells into the SG (Figure 3).

## 7. Conclusions

Alarmins initiate host defense against danger signals and are therefore suspected to play a pivotal role by linking innate and adaptive immune responses. Innate and adaptive immunity are central in the pathogenesis of pSS, as each of them represents a multi-step process leading to the triggering and perpetuation of disease. Further knowledge in deciphering and disheveling the role of alarmins in SS are required to expand our knowledge of the role of innate immune responses in the pathogenesis of this disabling disease, but also to provide the basis for new therapeutic strategies.

**Table 1 ijms-23-05671-t001:** A summary of alarmins linked to Sjögren’s syndrome and their biological effects.

Origin	Alarmins	Receptors	Biological Effects
Nuclear	HMBG-1	RAGE, TLR-4	↑ HMGB-1 in SG [19,20] and serum [20,21,22] ↑ sRAGE [22] Involvement of HMGB1/RAGE axis [22] and HMGB1/TLR4/NF-κB [24] Correlation with anti-SSA [21], ESSDAI [20], extra-glandular involvement [22]
	IL-1α	IL-1Ra	↑ IL-1*α* in saliva, tears, conjunctival epithelium, serum, SGEC [27,28,29,35,36] IL-1α in SG induce inflammation and SG dysfunction [26] ↓ IL-1Ra in saliva [31] Imbalance in salivary IL-1/IL-Ra associated with inflammation and SG dysfunction [33] Topical anti-IL-1 ↓ dry eye symptoms and corneal damage [27,39]
	IL-33	ST2, sST2	↑ IL-33 in blood, SGEC, tears [12,45,53,54] ↑ sST2 in blood [12,45] and correlate with ESSDAI [45] IL-33 stimulate IFN-γ secretion by NK and NK T cells [12] IFN-γ ↑ IL-33 production by SGEC [45] IL-33 and ST2 correlate with ILD [47]
Granule-derived	Defensins (α-defensin, β-defensin)	-	α-defensin-1 ↑ in tears, saliva and SG [177,178,179] α-defensin-1 ↑ type I IFN response genes [180] β-defensin-2-↑ in conjunctival epithelial cells, saliva [177,183] β-defensins 1 and 2 ↓ in SG [184]
Cytoplasmic	HSPs (HSP47, 60, 72 and 90)	TLR-2 and TLR-4	↑ HSP60 expression SG [65] ↑ HSP47 in blood [73] ↑ HSP72 and HSP90 in plasma of pSS with high fatigue scores [74]
	S100 proteins (S100A8/A9, S100B)	TLR-4, RAGE	↑ S100 A8/A9 in serum, SG, saliva, feces [87,88,89,90,92,93,94]. S100A8/A9 correlate with SG FS and MALT lymphoma [89,95] ↑ S100B in CSF [97,98,99] ↑ TLR-4 and RAGE in SGEC [102,103]
	ATP	P2X7R, P2Y2R	↑ P2X7R in SG, PBMC [109,110,113] ATP/P2X7R ↑ apoptosis of SGEC, NLRP3, IL-1β and IL-18 [111,112] P2X7R correlate with SG FS [109,110] and with MALT lymphoma [110] ↑ P2Y2R in SG [118,119]
	Uric acid	-	↑ in saliva and serum [130] ↓ in saliva [131]
	Gal-3	-	↑ in serum [142] ↑ Th17 and IL-17 [142]
	Calreticulin	-	Bind Ro peptide and ↑ autoimmune response against Ro60 [144,145,146,147]
	Tβ4	-	↑ in saliva [154]
	RNase 7		↑ SG [162]
	LL37	TLR-9	↑ SG [166] stabilize apoptotic DNA and activate maturation of B-cells via TLR-9 [166]
Plasma membrane	Annexins (annexin A2)	-	↑ in SG and saliva [186,187] Correlation with MALT [186]
ECM	Decorins	TLR-2 and TLR-4	Signal through TLR-4 [196]. Induction of TNF-α in splenic tissue [196]

## Figures and Tables

**Figure 1 ijms-23-05671-f001:**
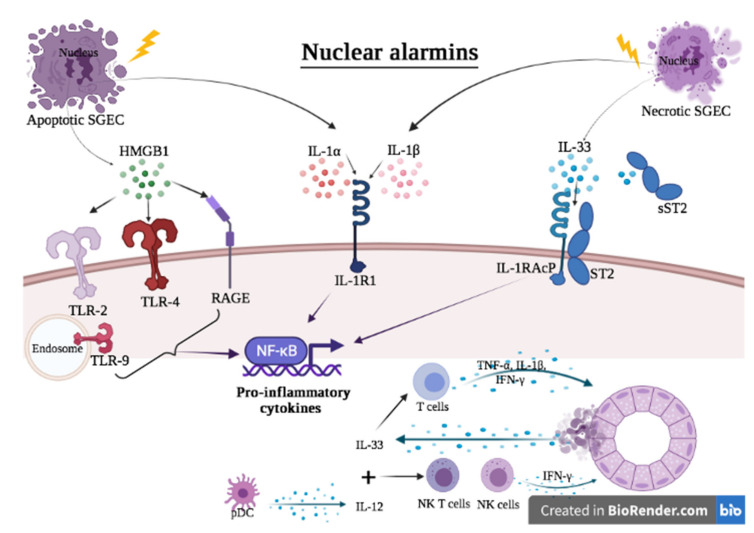
Hypothesis of the contribution of nuclear alarmins to the pathogenesis of Sjögren’s syndrome. Apoptotic and necrotic SGEC are a potent source of nuclear alarmins. Once in the extracellular milieu, they bind to their receptor, leading to the secretion of pro-inflammatory cytokines following the activation of intracellular pathways such as NF-κB. IL-33 works in concert with IL-12 to activate NK and NKT cells, thus promoting the release of IFN-γ. IFN-γ in turn acts on SGEC to further release IL-33 in the extracellular milieu, thereby constituting a vicious auto-inflammatory loop.

**Figure 2 ijms-23-05671-f002:**
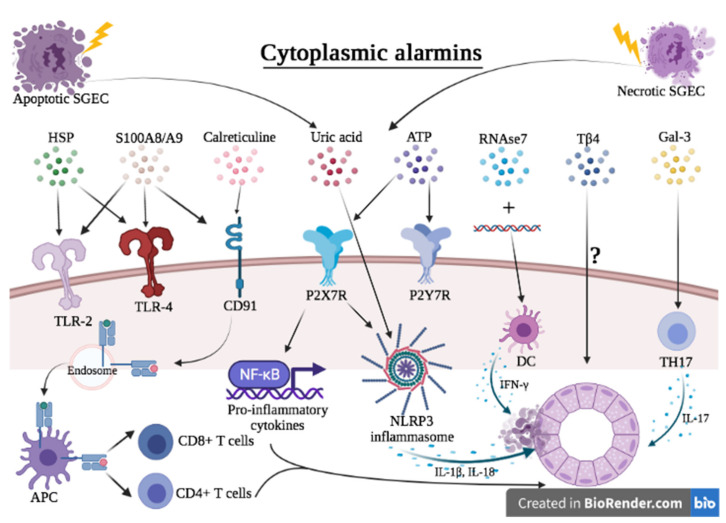
Hypothesis of the contribution of extracellular alarmins to the pathogenesis of Sjögren’s syndrome. Apoptotic and necrotic SGEC are a potent source of extracellular alarmins. Once in the extracellular milieu, they bind to their receptor, leading to the secretion of pro-inflammatory cytokines through the activation of intracellular pathways such as NF-κB or via the activation of NLRP3 inflammasome. DC are activated and are a potent source of IFN-γ. IFN-γ in turn acts on SGEC to further release alarmins in the extracellular milieu, thereby constituting a vicious auto-inflammatory loop. HSP can also be captured by surrounding APC, allowing cross-presentation of the pept i confirm ides by MHC I that further contribute to progression toward autoimmunity by activating CD8+ and CD4+ T cells.

**Figure 3 ijms-23-05671-f003:**
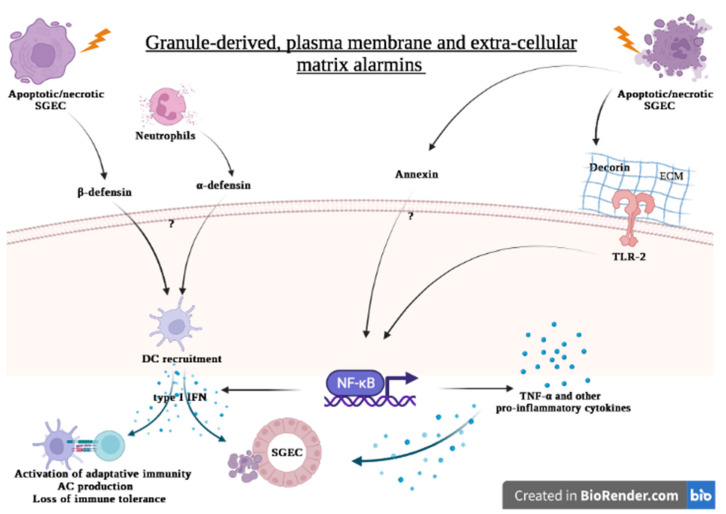
Hypothesis of the contribution of granule-derived, plasma membrane and extra-cellular matrix alarmins to the pathogenesis of Sjögren’s syndrome. Apoptotic and necrotic SGEC are a potent source of granule-derived, plasma membrane and extra-cellular matrix alarmins. Neutrophils contribute to the release of α-defensin. Once in the extracellular milieu, they bind to their receptor, leading to the secretion of pro-inflammatory cytokines through the activation of intracellular pathways such as NF-κB. DC are recruited and activated by α and β-defensin, leading in type I IFN secretion. IFN-γ acts on SGEC to further release alarmins in the extracellular milieu, thereby constituting a vicious auto-inflammatory loop. DC activation further contributes to the progression toward autoimmunity by activating the adaptive immune system.

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
