# Peer review of "The Involvement of Alarmins in the Pathogenesis of Sjögren’s Syndrome"

_ijms, 2022, doi:10.3390/ijms23105671_

Round 1
Reviewer 1 Report
//
This mauscript, a review by Dr. Sarrand et al. that provides an extensive overview of the current literature, addresses the potential roles that various molecular factors defined as ¨alarmins¨might play in the pathology of Sjogren´s Syndrome (SS). The presentation is divided into a discussion of alarmins based on cellular compartmentation. Although not directly pointed out in the paper, it appears that a common element in immune activations involves the NF-kb pathway…a pathway identified as critical in SS disease development. Overall, the manuscript is a comprehensive description of the topic that should be a nice reference for the field. However, a couple of issues should be addressed:
- The Introductory paragraph requires a better description of what exactly is an alarmin, especially in light of its long-term evolutionary changes in the field. This point is clearly evident in reading the third paragraph of the Introduction. The sentence (lines 59-60) is totally unclear, furthermore it suggests that SS is a microbial-induced disease, which is still debatable.
- The last two sentences of the Introduction suggests that this review focuses on the role of different types of alarmins in the pathogenesis of SS, but really is a review of various alarmins that have been reported to be dysregulated in SS and what/how they may contribute to or correlate with the SS pathology.
- Although there are two figures shown in the text, neither is actually referenced in the text and neither has a figure legend. Furthermore, is there a reason not to present a similar figure for annexins and defensins?
- While the manuscript is well-written, the verb usage requires review. Multiple times the verbs are wrongly singular vs plural.
Author Response
Dear Reviewer,Â
Thank you for reviewing our manuscript .Â
- we have accordingly updated the introductory section on alarmins.
- The text has been modified accordingly to review.. and figures have been referenced throughout the text.Â
- We have added new figures including defenses and decorins.
Thank you again for your review.
Kind regards
Â
Reviewer 2 Report
It is a lack of basic introduction to primary Sjögren's syndrome. Primary SS is related to multiple factors. The genetic locus most significantly associated with primary SS is the major histocompatibility complex/human leukocyte antigen region. Environmental factors, such as glandular viral infection, could prompt epithelial cells to activate the HLA-independent innate immune system through toll-like receptors. Epithelial cells in Sjögren's syndrome lesions are active participants in the induction and perpetuation of the inflammatory process. Typical SS ANA patterns are SSA/Ro and SSB/La, of which anti-SSB/La is far more specific; anti-SSA/Ro is associated with numerous other autoimmune conditions. Dysregulation of apoptosis (programmed cell death) is believed to play a role in the pathogenesis of a variety of autoimmune diseases,
The information is scattered and not well understood. It needs an overall graph about alarmins pathway and multifactorial interactions about pSS.
Author Response
Dear reviewer,Â
Â
The work is scaffolded following extra cellular and intra cellular alarmins . I am confused when you underline that it is scattered and not well understood. I am very sorry for that .Â
We have included new figures and added a graph as demanded
Kind regards
Â